# Is paternal age associated with transfer day, developmental stage, morphology, and initial hCG-rise of the competent blastocyst leading to live birth? A multicenter cohort study

**Maria Buhl Borgstrøm**[1,2]*, Marie Louise Grøndahl[1], Tobias W. Klausen[3], Anne K. Danielsen[4,5], Thordis Thomsen[5,6], Ursula Bentin-Ley[7], Ulla B. Knudsen[8,9], Steen Laursen[10], Morten R. Petersen[11], Katrine Haahr[12], Karsten Petersen[13], Josephine G. Lemmen[14], Johnny Hindkjær[15], John Kirk[16], Jens Fedder[17], Gitte J. Almind[18], Christina Hnida[19], Bettina Troest[2,20], Betina B. Povlsen[21], Anne Zedeler[22], Anette Gabrielsen[8], Thomas Larsen[23], Ulrik S. Kesmodel[2,20]

1 Department of Obstetrics and Gynecology, The Fertility Clinic, Copenhagen University Hospital Herlev, Herlev, Denmark, 2 Aalborg University, Aalborg, Denmark, 3 Department of Hematology, Copenhagen University Hospital Herlev, Herlev, Denmark, 4 Department of Gastroenterology, Copenhagen University Hospital Herlev, Herlev, Denmark, 5 Department of Clinical Medicine, University of Copenhagen, Copenhagen, Denmark, 6 Department of Anaesthesiology, Copenhagen University Hospital Herlev, Herlev, Denmark, 7 Danish Fertility Clinic, The Fertility Partnership Denmark, Copenhagen, Denmark, 8 The Fertility Clinic at Horsens regional hospital, Horsens, Deenmark, 9 Department of Clinical Medicine, Aarhus University, Aarhus, Denmark, 10 The Fertility Clinic IVF-syd, Fredericia, Denmark, 11 Department of Obstetrics and Gynecology, The Fertility Clinic, Copenhagen University hospital, Rigshospitalet, Denmark, 12 Stork IVF Clinic, Copenhagen, Denmark, 13 VivaNeo Ciconia Fertility Clinic, Højbjerg, Denmark, 14 Vitanova Fertility Center, København, Denmark, 15 Aagaard Fertility Clinic, Aarhus, Denmark, 16 Maigaard Fertility Clinic, Aarhus, Denmark, 17 The Fertility Clinic, Odense University Hospital, Odense, Denmark, 18 Copenhagen Fertility Center, Copenhagen, Denmark, 19 The Fertility Clinic, Zealand University Hospital Køge, Køge, Denmark, 20 The Fertility Unit, Aalborg University Hospital, Aalborg, Denmark, 21 The Fertility Clinic, Skive Regional Hospital, Skive, Denmark, 22 Department of Obstetrics and Gynecology, The Fertility Clinic, Copenhagen University Hospital, Hvidovre, Denmark, 23 Danish Medical Data Center, Copenhagen, Denmark

* maria.borgstroem@regionh.dk

**Data Availability Statement:** To adhere with GDPR rules and regulations data were pseudonymous,

## Abstract

In this study we investigated whether age of men undergoing assisted reproductive technology (ART) treatment was associated with day of transfer, stage, morphology, and initial hCG-rise of the competent blastocyst leading to a live birth? The design was a multicenter historical cohort study based on exposure (age) and outcome data (blastocyst stage and morphology and initial hCG-rise) from men whose partner underwent single blastocyst transfer resulting in singleton pregnancy/birth. The ART treatments were carried out at sixteen private and university-based public fertility clinics. We included 7246 men and women, who between 2014 and 2018 underwent controlled ovarian stimulation (COS) or Frozen-thawed Embryo Transfer (FET) with a single blastocyst transfer resulting in singleton pregnancy were identified. 4842 men with a partner giving birth were included, by linking data to the Danish Medical Birth Registry. We showed that the adjusted association between paternal age and transfer day in COS treatments was OR 1.06, 95% CI (1.00;1.13). Meaning that for every increase of one year, men had a 6% increased probability that the competent

which means that data were not directly personally identifiable (i.e. the CPR numbers were encrypted). Danish legislation allows sharing of pseudonymized data only if relevant legal permissions from e.g. the Danish Health Data Authority (Contact to Danish Health Data Authority: Cru-fp-vfd@regionh.dk), and the Danish Data Protection Agency (Contact to Danish Data Protection Agency: kontakt@sundhedsdata.dk) are obtained by the researchers who want the data to be made available.

**Funding:** MLG reports unsrestricted grants from Gedeon Richter, Nordic. The funders had no role in study design, data collection and analysis, decision to publish, or preparation of the manuscript.

**Competing interests:** I have read the journal's policy and the authors of this manuscript have the following competing interests: [Dr. Grøndahl reports unrestricted grants from Gedeon Richter, Nordic, during the conduct of the study. DR. Schiøler Kesmodel reports personal fees from Merck, personal fees from IBSA Nordic, outside the submitted work.]

blastocyst was transferred on day 6 compared to day 5. Further we showed that the mean difference in hCG values when comparing paternal age group 30–34, 35–39 and 40–45 with the age group 25–29 in those receiving COS treatment, all showed significantly lower adjusted values for older men. In conclusion we hypothesize that the later transfer (day 6) in female partners of older men may be due to longer time spent by the oocyte to repair fragmented DNA of the sperm cells, which should be a focus of future research in men.

## Introduction

Delayed parenthood is a growing trend worldwide with a higher age at conception as a consequence. Until recently, most research within the area of reproductive health has focused on women. However, there is now a global call in reproductive medicine for research on men. The probability of giving birth to a child, by natural conception or after assisted reproductive technology (ART) treatment, depends highly on the mother's age [1,2], but also the father's. Looking at national statistics, the mean age of fathers in Denmark has increased from 30.9 years in 1986 to 33.5 years in 2020 [3], while the mean age of fathers in the US has increased from 27.4 years in 1972 to 30.9 years in 2015 [4]. Advanced paternal age has been associated with reduced ability to conceive naturally [5] as well as by ART treatment. Following in vitro fertilization (IVF), reduced pregnancy- and live birth rates have been reported when the male partner is 40 years or older [6,7]. However, a recent review and meta-analysis based on the donor oocyte model suggests that advanced paternal age does not exert an independent effect on the outcome of ART cycles [8], while larger meta-analysis on naturally conceived pregnancies and miscarriage suggests a small independent effect of paternal age along with that of female age [9]. No association between blastocyst ploidy and paternal age has been found [10,11].

As soon as the sperm has fertilized the oocyte, the embryo development begins. This development is characterized by a collaboration between the sperm and the oocyte and the embryonic genome activation is under both male and female influence [12,13]. At blastulation, the cells differentiate into inner cell mass (ICM) and trophectoderm (TE) and the stage of a full blastocyst is reached on day 5 or 6 after fertilization. When a competent blastocyst meets a receptive endometrium and implants, a positive serum human chorionic gonadotropin (hCG) test will confirm the pregnancy, and a live birth will follow. In assisted reproductive technology (ART) treatment, cleavage stage embryo transfer at day 2 or 3 after fertilization has previously been practiced, while transfer at the blastocyst stage is now widely used, as the prevalence of aneuploidy in blastocysts is lower than in cleavage stage embryos [14,15]. To ensure selection of the best blastocyst to transfer, the timing, stage of development, and morphology are used as markers for competence. Worldwide, routine assessment involves grading of the development stage, the ICM, and TE as defined in the Istanbul consensus [16] and by Gardner and Schoolcraft [17]. The focus of the present study is the characteristics of the competent *in vitro* cultured blastocyst leading to a live birth in relation to the age of men.

The research in men within reproductive health is scarce. Furthermore, the age of men undergoing ART is increasing. For this reason, the potential effect of paternal age is important to investigate. We have recently shown that for every one-year increase in women's age there was a 5% reduced probability that the competent blastocyst was in an advanced development stage at the time of transfer. Likewise, we have shown that the initial hCG-rise was associated with women's age with the youngest women having the lowest hCG levels [18]. It is unknown

whether the age of men undergoing ART treatment is associated with the timing (day 5 or 6), the development stage, morphology, and the early implantation of the competent blastocyst. The aim of this study was to explore whether paternal age is associated with day of transfer, developmental stage, morphology, and initial hCG-rise of the competent blastocyst leading to a live birth.

## Materials and methods

### Study design and participants

The study was a historical multicenter cohort study with an analysis of prospectively collected data from men and women undergoing ART treatment. Inclusion criteria were all controlled ovarian stimulation (COS) followed by fresh single blastocyst transfer with IVF or intracyto-plasmatic sperm injection (ICSI) and Frozen-thawed Embryo Transfer (FET) cycles with Single Embryo Transfer (SET) resulting in a single implantation and singleton live birth. Exclusion criteria were treatments using oocyte and semen donation, or pre-implantation genetic testing. To ensure independence of observations in the dataset, treatments resulting in twin births were excluded. Further, for treatments resulting in the birth of two or more children (COS + FET) during the period of 2014–2018, only the first birth was included. Sixteen private and public Danish fertility clinics contributed with data on treatments resulting in pregnancy, from the database: Danish Medical Data Center (DMDC). Information on live birth was obtained from the Danish Medical Birth Register (DMBR) [19]. Using the woman's unique personal identification number (CPR), data from DMDC and DMBR were linked. The same dataset was used in our cohort study of the association between female age and the competent blastocyst [18]. The study is reported according to; Strengthening the Reporting of Observational Studies in Epidemiology (STROBE) [20].

### Treatment regimen

COS were performed with either long agonist or short antagonist protocols with either recombinant follicle-stimulating hormone or menotropin. When the leading two follicles reached 17-18mm final maturation and ovulation were triggered with hCG or gonadotropin-releasing hormone agonist administration. Oocyte pick-up (OPU) was performed 34–36 hours later. Women with regular menstrual cycles underwent modified natural FET cycles using hCG to induce ovulation and blastocyst transfer after 7 days. Other women underwent substituted FET cycles using estrogen to ensure a suitable endometrial thickness prepared for supplementation with progesterone and blastocyst transfer after 5–6 days. Within clinics, the *in vitro* procedures as well as the administration of progesterone and the hCG analysis were identical across age groups.

### Laboratory procedures

All centers used the blastocyst Gardner scoring system [17] as described in the Istanbul consensus, 2011 [16]. The laboratories at the fertility clinics were all following state of art techniques for IVF, ICSI, culture, evaluation and cryopreservation and are certified EU tissue directive centers. The primary reason to culture to day 6 before transfer or cryopreservation (main practice) was that preferred usable stage or morphology were not reached day 5 (i.e. if stage: <3 or TE<B). For ICSI, only MII oocytes were used for the treatments. Vitrification was used for cryopreservation of the blastocysts and subsequently, warming for the FET treatments.

## Study variables

**Exposure.** The age of every man at the date for OPU (age in whole years), was obtained from the clinical database DMDC and treated both as a continuous and a categorized variable. Six categories were used (21–24; 25–29; 30–34; 35–39; 40–45, $\geq$46 (maximum 70 years).

**Outcomes.** Information on timing, development stage, morphology and initial hCG-rise was obtained from the database DMDC. For COS, the evaluation of the blastocyst was done before transfer (day 5 or day 6 after fertilization) and for FET the evaluation was done before cryopreservation (day 5 or day 6 after fertilization). Further, the stage of the blastocyst, defined as the state of expansion and hatching at the day of blastocyst transfer was categorized as stage 1–6; 1: Early blastocyst (blastocoele less than half the size of the embryo); 2: Blastocyst (blasto-coele half the size of the embryo or more); 3: Full blastocyst (blastocoele fills the entire embryo); 4: Expanded blastocyst (blastocoele larger than the full blastocyst and with a diminishing zona); 5: Hatching blastocyst (TE starts to break through the zona); 6: Fully hatched blastocyst (blastocyst has left the zona). Morphology parameters for blastocysts with a stage score between 3–6, including an assessment of the ICM and TE categorized as A, B or C. ICM was evaluated as: A: Many cells and closely packed; B: Several cells and clustered loosely; C: Limited number of cells. Looking at TE, these were assessed as: A: Many cells creating an organized epithelium; B: Few cells establishing a loose epithelium; C: Limited large cells [17]. In addition, the stage and morphological parameters were categorized as; Score Group 1 (highest): 6AA, 6BA, 5AA, 5BA, 4AA, 4BA, Score Group 2: 6AB, 6BB, 6CB, 6CA, 5AB, 5BB, 5CB, 5CA, 4AB, 4BB, 4CB, 4CA and Score Group 3 (lowest): 6AC, 6BC, 6CC, 5AC, 5BC, 5CC, 4AC, 4BC, 4CC, 3AA, 3AB, 3BA, 3AC, 3CA, 3BB, 3BC, 3CB, 3CC and early blastocyst. The grouping was based primarily upon high score in stage and TE as suggested in several studies [21,22]. To ensure uniformity regarding the evaluation of the blastocysts in the participating clinics, the embryologists receive annual internal training. The multivariable analyses were adjusted for clinic, as some variation may exist between IVF laboratories.

The hCG level at the time of implantation was measured as a continuous variable and defined as the initial serum hCG value (IU/L). As hCG was measured at different time points, all values of hCG were standardized to 11 days after SET. The standardization was performed by creating a linear regression analysis with day of measure as independent variable, and log hCG as dependent variable. Using estimates from the regression analysis on expected change of hCG values, each value was transformed to an expected hCG at day 11.

**Covariates.** Using information from the database DMDC and DMBR [19], the following covariates were included in the statistical models; Female age on the day of OPU (years, continuous); Female cigarette smoking categorized as: non-smoker, quit smoking during pregnancy, and smoker; Female Body Mass Index (BMI) (kg/m$^2$, continuous); diagnosis categorized as: anovulation, tube factor, endometriosis, male factor, unexplained, and other; and clinic categorized as 1–16.

## Statistical methods

**Missing information.** Only competent blastocysts resulting in live births were included in the cohort Incomplete cases were removed from the analyses.

**Data analyses.** Continuous and categorical variables were compared by t-test or one-way ANOVA, respectively. The association between paternal age and transfer day/cryopreservation day was tested with logistic regression. The association between paternal age and morphology was tested with ordinal logistic regression. For these analyses we recoded the morphology parameters of the TE and ICM and category C was used as reference. The association between paternal age and the hCG level at the time of implantation was tested with linear regression

and multivariable linear regression. All analyses for paternal age were stratified by type of treatment (COS or FET). We present results from both crude and adjusted analyses, adjusting for female age, female BMI, female smoking, diagnosis and clinic [23,24]. Interaction between male age and female age (both continuous) was tested by including the interaction term male age * female age in the adjusted analyses of COS and FET combined.

The covariates chosen for adjustment were similar to those in our previous paper [18]. It could be argued that male BMI and male smoking would be obvious for adjustment, but male BMI and male smoking had a relative high number of missing values in our dataset (BMI 41%, smoking 24%). To test the sensitivity, we exchanged female BMI and female smoking for male BMI and male smoking in the adjusted analysis, since adding male BMI and male smoking to the list of covariates would potentially have given us problems with collinearity. Further sensitivity analyses were performed with inclusion of total FSH dose, fertilization method, and sex of the child for COS treatments and type of FET treatment for FET treatments. We performed other sensitivity analyzes by testing the day 5 blastocysts exclusively, we stratified the dataset further into IVF and ICSI and we tested the sensitivity by removing blastocysts in FET which were first transfers.

To investigate the assumptions for ordinal logistic regression each analysis was divided into several binary logistic regressions, representing each level of the dependent variable, including lower levels against higher levels. The odds ratios from these models and their 95% confidence intervals were compared to the odds ratios from the ordinal logistic regression model. For linear regression analyses, assumptions were assessed using residuals plots. For all multivariable analyses', collinearity was investigated. All statistical tests were two-sided and P-values $<0.05$ were considered statistically significant. Data analysis was performed with the statistical software R, version 3.4.1 [25].

**Ethical approval.**   Due to the historical design of the study, written informed consents were not possible to obtain from the participants. Accordingly, an ethical approval was given by the Danish Patient Safety Authority (Project number: 3-3013-2604/1). Further an ethical approval for this study was given by the Danish Data Protection Agency, Capital Region, Denmark (j.nr.: VD-2018-282, I-Suite nr.: 6522). The Danish Health Data Authority (FSEID-00003760) accepted and approved access to the DMBR and to the research platform where the statistics were carried out. To adhere with GDPR rules and regulations data were pseudonymous, which means that data were not directly personally identifiable (i.e. the CPR numbers were encrypted). Danish legislation allows sharing of pseudonymized data only if relevant legal permissions from e.g. the Danish Health Data Authority, and the Danish Data Protection Agency are obtained by the party who wants the data to be made available, along with separate permission to be obtained by the authors to hand over the data to a third party.

## Results

This cohort included data from 7246 men and couples participating in ART treatment with a transferred blastocyst resulting in a clinical pregnancy. In total, 4842 competent blastocysts developed into a live birth and were included in the analyses (S1 Fig). A total of 2044 men participated in a COS treatment, and 2798 participated in FET. Mean age for men in COS/FET were 34.7/34.5 years (Table 1). Mean BMI for COS/FET were 26.2/26.0. Most men were non-smokers: 82.1%/85.0%. Regarding the blastocyst characteristics for COS/FET 59.3%/50.4% were transferred on development stage 4, 66.4%/56.5% were TE grade A and 68.6%/59.2% were ICM grade A. Mean hCG for COS/FET were respectively 320IU/L and 498IU/L (Table 1). 35.8% of the men were between 30–34 year, and 27.2% were between 35–39 year (S1

**Table 1. Men, treatment, blastocyst timing, morphology and implantation characteristics.**

| Variable | Men–COS[1] | Men–FET[2] |
|---|---|---|
| **Age** * **(year), mean (sd)** | 34.7 (5.9) | 34.5 (6.1) |
| **Age** * **(year), range** | 21–68 | 19–68 |
| **Age** * **(year–category), n (%)** | | |
| total | 2044 (42.2) | 2798 (57.8) |
| 21–24 | 28 (1.4) | 52 (1.8) |
| 25–29 | 329 (16.1) | 542 (18.7) |
| 30–34 | 746 (36.5) | 987 (35.3) |
| 35–39 | 594 (29.1) | 721 (25.8) |
| 40–45 | 252 (12.3) | 383 (13.7) |
| 46–70 | 95 (4.6) | 131 (4.7) |
| **BMI (kg/m$^2$), mean (sd)** | 26.2 (3.9) | 26.0 (3.8) |
| missing | 851 | 1125 |
| **Smoking, cigarettes/day, n (%)** | | |
| 0 | 1260 (82.1) | 1808 (85) |
| 1–5 | 92 (6.0) | 100 (4.7) |
| 6–10 | 83 (5.4) | 98 (4.6) |
| 11–20 | 92 (6.0) | 110 (5.2) |
| >20 | 8 (0.5) | 11 (0.5) |
| missing | 509 | 671 |
| **Female partners parity, n (%)** | | |
| 1 | 1364 (72.9) | 1776 (71.2) |
| 2 | 440 (23.5) | 605 (24.3) |
| ≥3 | 68 (3.6) | 112 (4.5) |
| missing | 172 | 305 |
| **Indication for cause of infertility, n (%)** | | |
| anovulation | 237 (11.8) | 321 (11.7) |
| tube factor | 154 (7.7) | 251 (9.1) |
| endometriosis | 92 (4.6) | 132 (4.8) |
| male factor | 716 (35.5) | 984 (35.7) |
| Unexplained[3] | 686 (34.1) | 821 (29.8) |
| Other[4] | 127 (6.3) | 244 (8.9) |
| missing | 32 | 45 |
| **Number of previous transfers, n (%)** | | |
| 0 | 1354 (66.2) | 526 (18.8) |
| 1 | 295 (14.5) | 1035 (37.0) |
| 2 | 171 (8.4) | 522 (18.7) |
| 3 | 109 (5.3) | 289 (10.3) |
| ≥4 | 115 (5.6) | 426 (15.2) |
| **Fertilization method, n (%)** | | |
| IVF[5] | 998 (48.8) | 1208 (43.2) |
| ICSI[6] | 1046 (51.2) | 1590 (56.8) |
| **Timing, n (%)** | | |
| day 5 blastocyst[7] | 2001 (97.9) | 2181 (77.9) |
| day 6 blastocyst[8] | 43 (2.1) | 617 (22.1) |
| **Stage, n (%)** | | |
| 1[9] | 33 (1.6) | 20 (0.7) |
| 2[9] | 32 (1.5) | 12 (0.4) |

(*Continued*)

**Table 1.** (Continued)

| Variable | Men–COS[1] | Men–FET[2] |
|---|---|---|
| 3 | 289 (14.1) | 465 (16.6) |
| 4 | 1212 (59.3) | 1409 (50.4) |
| 5 | 461 (22.7) | 738 (26.4) |
| 6 | 17 (0.8) | 154 (5.5) |
| **Trophectoderm (TE), n (%)** | | |
| A | 1301 (66.4) | 1489 (56.5) |
| B | 593 (30.3) | 1069 (40.5) |
| C | 65 (3.3) | 79 (3.0) |
| missing | 85 | 161 |
| **Inner cell mass (ICM), n (%)** | | |
| A | 1343 (68.6) | 1561 (59.2) |
| B | 561 (28.6) | 1013 (38.4) |
| C | 55 (2.8) | 63 (2.4) |
| missing | 85 | 161 |
| **hCG[10], mean (sd)** | 341 (194) | 457 (286) |
| missing | 320 | 498 |

*Paternal age at oocyte pick up

[1] COS: Controlled Ovarian Stimulation

[2] FET: Frozen-thawed Embryo Transfer

[3]Unexplained: Couples with unexplained infertility, [4]Other: Female infertility caused by conditions in ovary, uterus, cervix or caused by other conditions (hepatitis, habitual abortion, asymptomatic HIV), [5]IVF: In Vitro Fertilization, [6]ICSI: Intracytoplasmic Sperm Injection

[7]Day 5 blastocyst: For COS defined as blastocyst transfer at day 5 and for FET defined as cryopreservation at day 5

[8]Day 6 blastocyst: For COS defined as blastocyst transfer at day 6 and For FET defined as cryopreservation at day 6

[9]Stage 1 and 2 (Group 3) are included in the group analysis only

[10]First measurement of serum human chorionic gonadotrophin (hCG).

Table). Further stratification on paternal age can be seen in S1 Table. The squared correlation coefficient between women's- and men's age was $R^2 = 0.35$ (S2 Fig).

## Association between paternal age and stage and morphology of the competent blastocyst

In S2 and S3 Tables, men's unadjusted mean age for COS and FET according to stage and morphology is shown. In FET, the age differed significantly for TE and Score Group (S3 Table), but the absolute differences were small. No statistically significant or clinically relevant differences were seen for COS (S2 Table).

The adjusted association between age and transfer day in COS treatments showed that for every increase of one year, a man had a 6% increased probability that the competent blastocyst was transferred at day 6 compared to day 5 (OR 1.06, 95%CI (1.00;1.13)) (Table 2). For FET, while the crude analysis of TE, ICM and Score Group showed significant associations with paternal age, no statistically significant associations were observed after adjustment (Table 2). A sensitivity analyses for COS treatments further adjusting for total FSH dose, fertilization method and sex of the child resulted in a stronger association between paternal age and transfer day (OR 1.11, 95% CI 1.02;1.20) and otherwise comparable results (S4 Table). Further, for FET treatments, when adding type of FET treatment to the adjusted analyses the results was comparable (S5 Table).

**Table 2. The association of paternal age\* with day of transfer, developmental stage and morphology of the competent blastocyst after COS[1] and [2]FET.**

| COS | N | OR | OR-adjusted\*\* |
|---|---|---|---|
| **Age**\* | **2044** | | |
| **Transfer day** | 2044 | | |
| 5 | 2001 | ref. | ref. |
| 6 | 43 | 1.05 (1.01;1.10) | 1.06 (1.00;1.13) |
| **Stage (3–6)** | 1979 | 0.99 (0.98;1.01) | 1.01 (0.99;1.03) |
| missing | 65 | | |
| **TE[3] (A-C)** | 1959 | 1.00 (0.99;1.02) | 1.00 (0.98;1.02) |
| missing | 85 | | |
| **ICM[4] (A-C)** | 1959 | 1.00 (0.99;1.02) | 0.99 (0.98;1.02) |
| missing | 85 | | |
| **Group (1–3)** | 1959 | 0.99 (0.98;1.01) | 0.99 (0.98;1.02) |
| missing | 85 | | |
| **FET** | | | |
| **Age**\* | 2798 | | |
| **Cryopreservation day** | 2798 | | |
| 5 | 2181 | ref. | ref. |
| 6 | 617 | 1.02 (1.00;1.03) | 1.01 (0.98;1.03) |
| **Stage (3–6)** | 2766 | 1.00 (0.99;1.01) | 0.99 (0.98;1.01) |
| missing | 32 | | |
| **TE[3] (A-C)** | 2637 | 1.02 (1.01;1.03) | 0.99 (0.98;1.01) |
| missing | 161 | | |
| **ICM[4] (A-C)** | 2637 | 1.01 (1.00;1.03) | 0.99 (0.98;1.01) |
| missing | 161 | | |
| **Group (1–3)** | 2637 | 1.01 (1.00;1.02) | 0.99 (0.98;1.01) |
| missing | 161 | | |

Logistic regression. Multivariable logistic regression. Ordinal logistic regression. Ordinal multivariable logistic regression.

\*Paternal age at oocyte pick up

\*\*Adjusted for female age, female BMI, female smoking, diagnosis and clinic

[1]COS: Controlled Ovarian Stimulation

[2]FET: Frozen-thawed Embryo Transfer

[3]TE: Trophectoderm

[4]ICM: Inner Cell Mass.

## Association between paternal age and the hCG level at implantation of the competent blastocyst

Women with men of advanced age (≥46) in both COS and FET had the highest unadjusted mean hCG values 370.3/510.8 (Table 3). However, the adjusted mean difference of hCG values when comparing age group 30–34, 35–39 and 40–45 with the age group 25–29 in those receiving COS treatment, all showed significantly lower adjusted values for older men (-51.7, 95% CI (-80.5;-23.0)), (-38.8, 95% CI (-72.6;-5.0)), (-55.2, 95% CI (-96.6; -13.8)) (Table 3). In those receiving FET, the crude mean difference was significantly higher only for men aged ≥46 compared to 25–29 (61.7, 95% CI (1.1;122.2)), but the association was not significant in the adjusted analysis (Table 3).

For FET, 526 women had previously not had a transfer with fresh blastocyst (all blastocysts were frozen in the COS treatment) (Table 2) and hence the first transfer was a FET. We did a re-analysis without these treatments, which did not change the conclusions (S6 Table). In

**Table 3. The association of paternal age* with initial hCG[1] rise at implantation, of the competent blastocyst after COS[2] and FET[3].**

| COS Paternal age* | N | Missing | Mean hCG[2] (sd) | Meandiff. (95%CI) | P-value | Adj. meandiff. (95%CI)** | P-adj |
|---|---|---|---|---|---|---|---|
| 18–24 | 28 | 0 | 344.9 (156.7) | -22.3 (-97.5;52.9) | 0.56 | -9.2 (-86.5;68.1) | 0.82 |
| 25–29 | 289 | 40 | 367.2 (197.6) | ref. | | ref. | |
| 30–34 | 631 | 115 | 328.9 (184.1) | -38.3 (-65.3;-11.3) | 0.01 | -51.7 (-80.5;-23.0) | <0.001 |
| 35–39 | 492 | 102 | 341.2 (206.9) | -26.0 (-54.1;2.2) | 0.07 | -38.8 (-72.6;-5.0) | 0.02 |
| 40–45 | 204 | 48 | 331.5 (172.4) | -35.7 (-70.5;-1.0) | 0.04 | -55.2 (-96.6;-13.8) | 0.01 |
| 46–99 | 80 | 15 | 370.3 (228.5) | 3.1 (-44.9;51.1) | 0.90 | -17.3 (-70.6;36.0) | 0.52 |
| Total | 1724 | 320 | | | | | |
| P overall | | | | | 0.07 | | 0.01 |
| **FET Paternal age*** | | | | | | | |
| 18–24 | 46 | 6 | 420.7 (240.7) | -28.4 (-15.0;58.2) | 0.52 | -18.5 (-111.4;74.5) | 0.70 |
| 25–29 | 461 | 63 | 449.2 (277.1) | ref. | | ref. | |
| 30–34 | 839 | 148 | 465.2 (289.9) | 16.1 (-16.4;48.6) | 0.33 | 0.04 (-35.6;35.7) | 0.99 |
| 35–39 | 555 | 166 | 441.2 (278.7) | -8.0 (-43.2;27.3) | 0.66 | -11.4 (-54.5;31.6) | 0.60 |
| 40–45 | 294 | 89 | 465.7 (287.9) | 16.6 (-25.2;58.4) | 0.44 | -9.0 (-60.7;42.6) | 0.73 |
| 46–99 | 105 | 26 | 510.8 (331.5) | 61.7 (1.1;122.2) | 0.04 | 24.9 (-46.6;96.4) | 0.49 |
| Total | 2300 | 498 | | | | | |
| P overall | | | | | 0.19 | | 0.90 |

Linear regression. Multivariable linear regression.

*Paternal age at oocyte pick up

**Adjusted for female age, female BMI, female smoking, diagnosis and clinic

[1]human chorionic gonadotrophin

[2]COS: Controlled Ovarian Stimulation

[3]FET: Frozen-thawed Embryo Transfer.

relation to Tables 2 and 3 we performed a sensitivity analysis with only the day 5 blastocysts, which did not change the conclusions (S7–S10 Tables).

Adjusting analyses for male BMI and male smoking instead of female BMI and female smoking in the adjusted analysis in Tables 2 and 3 did not change the conclusions (S7–S10 Tables).

Further, the interaction between women's- and men's age in relation to stage, morphology and the hCG level at implantation was tested in the full dataset (COS+FET), and no interactions were found (S15 Table).

## Discussion

In this historical multicenter cohort study, we aimed to explore whether paternal age was associated with timing (transfer day (COS), cryopreservation day (FET)), stage, morphology, and the hCG level at implantation of the competent blastocyst. Our main findings were that in COS treatments, men's age was associated with transfer day of the competent blastocyst. Furthermore, reduced hCG-values were seen in the pregnant partner of men ≥30 years compared to 25–29 years in COS treatments. No significant associations were seen in FET treatments.

One main strength of this study is the high number of exposure and outcome data, which is due to the nationwide coverage of the databases. This resulted in narrow confidence intervals, which support that our estimates are close to the true value, and the level of random error is

low. Looking at the risk of systematic errors it is worth mentioning that due to reorganization of the database DMBR, it was not possible to have complete follow-up. From DMDC we had data from 2014–2018, and from DMBR we had data from 2014-March 2019. Missing data on live births throughout 2019 increased the risk of selection bias [26,27]. The subjective assessment of the blastocyst morphology is important to consider, as it could have introduced some degree of misclassification, i.e. information bias [28]. Because of this risk of variation between clinics in assessment of the blastocyst and potential other procedures, we adjusted for clinic in the analyses. The cohort was designed to include SET of blastocysts resulting in a single live birth. This ensured correct data as only one blastocyst was assessed and transferred. Further, by excluding twin births following SET we did not have to consider the influence of multiple implantations on hCG levels. On the other hand, having a design exclusively including blastocyst leading to a pregnancy and live birth will not present results differentiating between viable and not viable blastocysts, which could be perceived as a limitation. Studies investigating similar association as this study are limited, and hence knowledge of possible confounders is limited. Therefore, it cannot be ruled out that some residual confounding remains [23,24]. Sensitivity analyses were performed for consideration of different covariates. Finally, the exclusion of donor gametes limits the external validity to couples using their own gametes, but as this group constitutes the vast majority of men and women undergoing ART treatment this should be considered a minor issue.

For COS treatment, we found that for every increase of one year, a man had a 6% increased probability that the competent blastocyst was transferred at day 6, compared to day 5. While a 6% increase is not much per se, the 6% increased risk accumulates over a life span and may eventually add up to a considerably increased risk. This could indicate that by increasing paternal age, the competent blastocyst develops slower and hence reaches the full blastocyst stage for transfer later. A trend towards a slower development has earlier been suggested in a study on 1,023 oocyte donor cycles reporting significantly fewer embryos reaching the blastocyst stage day 5 if the male partner was above 55 years [29]. Likewise, a recent study on 3,837 cycles investigating the interaction between maternal and paternal age on morphological parameters showed that blastocyst development was negatively influenced by increasing maternal and paternal age [30]. However, a subsequent meta-analysis representing 12,538 oocyte-donation cycles failed to show any effect of paternal age on embryo development *in vitro* [31]. Further, our result should be interpreted with caution. First, OR's overestimate the true risk ratio, especially in situations where the outcome occurs frequently. Second, there is always a risk of type 1 error, and the risk increases with increasing number of statistical tests performed. However, we followed a pre-planned analysis plan. In this study, we focused selectively on the competent blastocyst reaching a live birth, and hence we do not know if paternal age had a general impact on assessment scores of the usable (transferred and cryopreserved) blastocysts in the participating clinics. If the observed influence of male age reflects changes in the time to reach transferable developmental stage in general, remains unanswered.

Paternal genetic and epigenetic factors affecting embryo development and competence have been suggested [32]. The sperm methylome associates with the age of the man [33] and with increasing male age, an increased degree of sperm DNA damage has been observed [34]. As the mature spermatozoa has no/minimal DNA damage response [35], there will be an increasing task for the DNA repair capacity of the oocyte with increasing age of the man. Several studies, including autologous cycles, report a negative effect of paternal age in ART outcome when the female partner age is above 40 years, and it has been suggested that this could at least partly be explained by decreased oocyte competence to repair DNA breaks with increasing maternal age [36,37]. A new meta-analysis addressing the effect of paternal age on pregnancy loss suggests a small independent effect of paternal age along with that of female

age [9]. A possible synergism seems not to be present in the reported parameters in our study, as test for interaction between male age and female age was insignificant for all outcomes. We speculate, if the present observation on delayed blastocyst development with increasing paternal age resulting in transfer day 6, reflects an increase in cell cycle length due to time spend to repair fragmented DNA [38]. Even so, in contrast to transfer day reaching a statistically significant association with paternal age, no association with stage (also reflecting developmental speed) was present. This can potentially reflect that the delay is subtle.

In Denmark, ICSI is performed primarily in couples with severe male factor. As sperm quality has been suggested to influence blastocyst formation rate [39], we stratified by fertilization methods (IVF/ICSI) and no differences were observed between the two groups in blastocyst stage, morphology nor transfer day in relation to paternal age.

It is well known that a single hCG measurement is not an ideal marker of the quality of an implantation [40]. In the present study, the initial hCG-rise (one hCG measurement 11 days after transfer) reflects an excellent implantation as it sustained until the live birth and is used as a parameter to further characterize the competent blastocyst.

A slower blastocyst development is supported by the association we found between a lower initial hCG-rise with advancing paternal age following transfers in COS cycles, potentially reflecting a subtle postponement in implantation. Excluding the day 6 data from the analysis did not alter the lack of association between paternal age and stage and morphology, nor the association showing decrease in initial hCG with increasing paternal age, supporting an overall delay in implantation after COS by increasing paternal age. Interestingly, in our previous analyses of the association between female age and parameters related to the development speed of the competent blastocyst, we found that increasing female age was associated with significantly lower chance of the competent blastocyst being in the high developmental stage, while no significant association was present between female age and transfer day (day 5 or 6 after fertilization) [18]. With respect to initial hCG-rise after COS, we found systematically lower hCG-levels in young women aged 18–24 years [18], whereas the hCG-levels were significantly lower in women with older partners aged 30–45 years. While the association in women may reflect better embryo-endometrium synchrony in young women compared to older women [18], as suggested the association in men may well reflect the potentially later implantation.

In FET treatments, as in COS treatments, no associations were found between the stage and morphology of the competent blastocyst and paternal age, neither in the full data analysis nor in the analysis excluding blastocysts cryopreserved day 6. Same analyses for initial hCG-rise showed no associations to paternal age.

In the FET dataset, 526 were first-transfers and these blastocysts may represent another cohort of blastocysts than the blastocysts cryopreserved after transfer of the top blastocyst in the COS cycle. However, the sensitivity analysis without these 526 first transfers did not change the results. The difference in the association between initial hCG-rise and paternal age between COS and FET may reflect that the potential delay involves a blastocyst-endometrial asynchrony, which is less relevant in FET cycles.

In our previous analyses of the association between female age and parameters related to the competent blastocyst, we found similar results, showing no association between female age and cryopreservation day, stage, TE, ICM or group in FET cycles [18], suggesting these parameters to be completely independent of age of both partners. With respect to initial hCG-rise, we found systematically lower hCG-levels in young women [18] but no association with paternal age in FET cycles, which may reflect better embryo-endometrium synchrony in young women compared to older women [18], which would occur independently of paternal age.

The mean paternal age (at OPU) for COS and FET was 34.7/34.5 years, and the mean maternal age (at OPU) for COS and FET was respectively 32.0/32.2 years [18]. Even so, while

men were systematically older than women in our cohort, assessing the correlation between male and female age we found that age observations were evenly distributed around the regression line when plotting male against female age, suggesting that the age difference between men and women was fairly independent of young or older age.

In conclusion, our cohort data on competent blastocysts each reaching the birth of a child, demonstrate that advanced paternal age may be associated with increased time to reach the transferable stage and reduced level of hCG in the pregnant partner following COS and single blastocyst transfer. In FET treatments, none of the investigated associations reached statistical significance. We hypothesize 1) that the later transfer (day 6) in female partners of older men may be due to longer time spent by the oocyte to repair fragmented DNA of the sperm cells or an age dependent blastocyst-endometrial asynchrony, and 2) that the reduced level of hCG could reflect a delay in implantation which should be a focus of future research in men.

## Supporting information

**S1 Fig. Flow chart.** [1]Danish medical data center, [2]Preimplantation genetic testing, [3]Personal identification number, [4]Ultrasound testing for pregnancy, [5]Danish medical birth register. (DOCX)

**S2 Fig. The correlation between women's age and men's age at oocyte pick up.** (DOCX)

**S1 Table. Male age, treatment, blastocyst timing, morphology and implantation characteristics.** [*]Paternal age at oocyte pick up, [1]Unexplained: Couples with unexplained infertility, [2]Other: Female infertility caused by conditions in ovary, uterus, cervix or caused by other conditions (hepatitis, habitual abortion, asymptomatic HIV), [3]IVF: In Vitro Fertilization, [4]ICSI: Intracytoplasmic Sperm Injection, [5]Day 5 blastocyst: For COS defined as blastocyst transfer at day 5 and for FET defined as cryopreservation at day 5, [6]Day 6 blastocyst: For COS defined as blastocyst transfer at day 6 and For FET defined as cryopreservation at day 6, [7]First measurement of serum human chorionic gonadotrophin (hCG). (DOCX)

**S2 Table. The unadjusted association of paternal age[*] with day of transfer, developmental stage and morphology of the competent blastocyst after COS1 t–test.** One-way ANOVA. [*]Paternal age at oocyte pick up, [1]COS: Controlled Ovarian Stimulation, [2]TE: Trophectoderm, [3]ICM: Inner Cell Mass, [4]Group 1: 6AA, 6BA, 5AA, 5BA, 4AA, 4BA, Group 2: 6AB, 6BB, 6CB, 6CA, 5AB, 5BB, 5CB, 5CA, 4AB, 4BB, 4CB, 4CA, Group 3: 6AC, 6BC, 6CC, 5AC, 5BC, 5CC, 4AC, 4BC, 4CC, 3AA, 3AB, 3BA, 3AC, 3CA, 3BB, 3BC, 3CB, 3CC, 2AA, 2AB, 2BA, 2AC, 2CA, 2BB, 2BC, 2CB, 2CC, 1AA, 1AB, 1BA, 1AC, 1CA, 1BB, 1BC, 1CB, 1CC. (DOCX)

**S3 Table. The unadjusted association of paternal age[*] with day of transfer, developmental stage and morphology of the competent blastocyst after FET1 t–test.** One-way ANOVA. [*]Paternal age at oocyte pick up, [1]FET: Frozen-thawed Embryo Transfer, [2]TE: Trophectoderm, [3]ICM: Inner Cell Mass, [4]Group 1: 6AA, 6BA, 5AA, 5BA, 4AA, 4BA, Group 2: 6AB, 6BB, 6CB, 6CA, 5AB, 5BB, 5CB, 5CA, 4AB, 4BB, 4CB, 4CA, Group 3: 6AC, 6BC, 6CC, 5AC, 5BC, 5CC, 4AC, 4BC, 4CC, 3AA, 3AB, 3BA, 3AC, 3CA, 3BB, 3BC, 3CB, 3CC, 2AA, 2AB, 2BA, 2AC, 2CA, 2BB, 2BC, 2CB, 2CC, 1AA, 1AB, 1BA, 1AC, 1CA, 1BB, 1BC, 1CB, 1CC. (DOCX)

**S4 Table. The association of paternal age[*] with timing, stage and morphology of the competent blastocyst after COS[1].** Logistic regression. Multiple logistic regression. Ordinal logistic

regression. Ordinal multiple logistic regression. *Paternal age at oocyte pick up, **Adjusted for female age, female BMI, female smoking, diagnosis, clinic, FSH dose, fertilization method and sex of the child, 1COS: Controlled Ovarian Stimulation, 2TE: Trophectoderm, 3ICM: Inner Cell Mass.
(DOCX)

**S5 Table. The association of paternal age* with timing, stage and morphology of the competent blastocyst after FET[1].** Logistic regression. Multiple logistic regression. Ordinal logistic regression. Ordinal multiple logistic regression. *Paternal age at oocyte pick up, **Adjusted for female age, female BMI, female smoking, diagnosis, clinic and type of FET treatment, 1FET: Frozen-thawed Embryo Transfer, 2TE: Trophectoderm, 3ICM: Inner Cell Mass.
(DOCX)

**S6 Table. The association of men's age* with implantation, initial hCG[1] rise, of the competent blastocyst after FET[2] –without 526 first transfers.** Linear regression. Multiple linear regression. *Men's age at oocyte pick up, **Adjusted for female age, female BMI, female smoking, diagnosis and clinic, 1human chorionic gonadotrophin, 2FET: Frozen-thawed Embryo Transfer.
(DOCX)

**S7 Table. The association of men's age* with stage and morphology of the competent blastocyst after COS[1]—without 43 day 6 blastocysts.** Logistic regression. Multivariable logistic regression. Ordinal logistic regression. Ordinal multivariable logistic regression. *Men's age at oocyte pick up, **Adjusted for female age, female BMI, female smoking, diagnosis and clinic, 1COS: Controlled Ovarian Stimulation, 2TE: Trophectoderm, 3ICM: Inner Cell Mass.
(DOCX)

**S8 Table. The association of men's age* with stage and morphology of the competent blastocyst after FET[1] –without 617 day 6 blastocysts.** Logistic regression. Multivariable logistic regression. Ordinal logistic regression. Ordinal multivariable logistic regression. *Men's age at oocyte pick up, **Adjusted for male age, female BMI, female smoking, diagnosis and clinic, 1FET: Frozen-thawed Embryo Transfer, 2TE: Trophectoderm, 3ICM: Inner Cell Mass.
(DOCX)

**S9 Table. The association of men's age* with implantation, initial hCG[1] rise, of the competent blastocyst after COS[2]—without 43 day 6 blastocysts.** Linear regression. Multivariable linear regression. *Men's age at oocyte pick up, **Adjusted for female age, female BMI, female smoking, diagnosis and clinic, 1human chorionic gonadotrophin, 2COS: Controlled Ovarian Stimulation.
(DOCX)

**S10 Table. The association of men's age* with implantation, initial hCG[1] rise, of the competent blastocyst after FET[2]– without 617 day 6 blastocysts.** Linear regression. Multivariable linear regression. *Men's age at oocyte pick up, **Adjusted for female age, female BMI, female smoking, diagnosis and clinic, 1human chorionic gonadotrophin, 2FET: Frozen-thawed Embryo Transfer.
(DOCX)

**S11 Table. The association of men's age* with stage and morphology of the competent blastocyst after COS[1].** Logistic regression. Multivariable logistic regression. Ordinal logistic regression. Ordinal multivariable logistic regression. *Men's age at oocyte pick up, **Adjusted for female age, male BMI, male smoking, diagnosis and clinic, 1COS: Controlled Ovarian

Stimulation 2TE: Trophectoderm, 3ICM: Inner Cell Mass.
(DOCX)

**S12 Table. The association of men's age[*] with stage and morphology of the competent blastocyst after FET[1].** Logistic regression. Multiple logistic regression. Ordinal logistic regression. Ordinal multiple logistic regression. [*]Men's age at oocyte pick up, [**]Adjusted for female age, male BMI, male smoking, diagnosis and clinic, 1FET: Frozen-thawed Embryo Transfer, 2TE: Trophectoderm, 3ICM: Inner Cell Mass.
(DOCX)

**S13 Table. The association of men's age[*] with implantation, initial hCG[1] rise, of the competent blastocyst after COS[2].** Linear regression. Multiple linear regression. [*]Men's age at oocyte pick up, [**]Adjusted for female age, male BMI, male smoking, diagnosis and clinic, 1human chorionic gonadotrophin, 2COS: Controlled Ovarian Stimulation.
(DOCX)

**S14 Table. The association of men's age[*] with implantation, initial hCG[1] rise, of the competent blastocyst after FET[2].** Linear regression. Multiple linear regression. [*]Men's age at oocyte pick up, [**]Adjusted for female age, male BMI, male smoking, diagnosis and clinic, 1human chorionic gonadotrophin, 2FET: Frozen-thawed Embryo Transfer.
(DOCX)

**S15 Table. Analysis of interaction between women age (age) and male age (age_vir).**
(DOCX)

## Acknowledgments

We would like to thank all the participating fertility clinics contributing with data.

## Author Contributions

**Conceptualization:** Maria Buhl Borgstrøm, Marie Louise Grøndahl, Ulrik S. Kesmodel.

**Data curation:** Maria Buhl Borgstrøm, Marie Louise Grøndahl, Tobias W. Klausen, Ursula Bentin-Ley, Ulla B. Knudsen, Steen Laursen, Morten R. Petersen, Katrine Haahr, Karsten Petersen, Josephine G. Lemmen, Johnny Hindkjær, John Kirk, Jens Fedder, Gitte J. Almind, Christina Hnida, Bettina Troest, Betina B. Povlsen, Anne Zedeler, Anette Gabrielsen, Thomas Larsen, Ulrik S. Kesmodel.

**Formal analysis:** Maria Buhl Borgstrøm, Tobias W. Klausen, Anne K. Danielsen, Thordis Thomsen, Ulrik S. Kesmodel.

**Investigation:** Maria Buhl Borgstrøm, Marie Louise Grøndahl, Ulrik S. Kesmodel.

**Methodology:** Maria Buhl Borgstrøm, Ulrik S. Kesmodel.

**Project administration:** Maria Buhl Borgstrøm, Ulrik S. Kesmodel.

**Resources:** Maria Buhl Borgstrøm, Marie Louise Grøndahl.

**Software:** Maria Buhl Borgstrøm.

**Supervision:** Marie Louise Grøndahl, Ulrik S. Kesmodel.

**Writing – original draft:** Maria Buhl Borgstrøm, Marie Louise Grøndahl, Ulrik S. Kesmodel.

**Writing – review & editing:** Maria Buhl Borgstrøm, Marie Louise Grøndahl, Tobias W. Klausen, Anne K. Danielsen, Thordis Thomsen, Ursula Bentin-Ley, Ulla B. Knudsen, Steen

Laursen, Morten R. Petersen, Katrine Haahr, Karsten Petersen, Josephine G. Lemmen, Johnny Hindkjær, John Kirk, Jens Fedder, Gitte J. Almind, Christina Hnida, Bettina Troest, Betina B. Povlsen, Anne Zedeler, Anette Gabrielsen, Thomas Larsen, Ulrik S. Kesmodel.

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
