## [Decision Letter · Decision Letter 0]

13 Jan 2022

PONE-D-21-32713Is paternal age associated with transfer day, developmental stage, morphology, and initial hCG-rise of the competent blastocyst leading to live birth? a multicenter cohort studyPLOS ONE

Dear Dr. Borgstrøm,

Thank you for submitting your manuscript to PLOS ONE. After careful consideration, we feel that it has merit but does not fully meet PLOS ONE’s publication criteria as it currently stands. Therefore, we invite you to submit a revised version of the manuscript that addresses the points raised during the review process.

ACADEMIC EDITOR:The paper has been appreciated by the two Reviewers. However, there are some minor aspects that need to be addressed: 1. The Authors has underlined the importance of the results of the fresh transfer on day 6 compared to day 5. However, the number of transfers in day 6 was only 43. How can the authors be so sure not to have a type 1 error? The Authors may discuss this aspect. The manuscript should be also modified according to this possible bias. 2. The strategy to transfer a fresh embryo in day 6 is not so diffuse given the idea that in day 6 the endometrium is less receptive. As a matter of fact, the number of fresh transfers in day 6 is low. The Authors stated that they have corrected for centers but results are not shown. Is it possible that few centers adopted this strategy and they had older patients? 3. Was there some kind of standardization of blastocyst morphology among centers? 4. Given the small ORs, the conclusions seem overemphasized. 5. The choice of the bHCG levels is debatable. Please add some references to support this choice.

We look forward to receiving your revised manuscript.

Kind regards,

Paola Viganò

Academic Editor

PLOS ONE

Journal Requirements:

3. Thank you for submitting the above manuscript to PLOS ONE. During our internal evaluation of the manuscript, we found significant text overlap between your submission and the following previously published works, some of which you are an author.

https://bmcpregnancychildbirth.biomedcentral.com/articles/10.1186/s12884-020-03348-2

https://www.fertstert.org/article/S0015-0282(20)32191-9/fulltext

Please revise the manuscript to rephrase the duplicated text, cite your sources, and provide details as to how the current manuscript advances on previous work. Please note that further consideration is dependent on the submission of a manuscript that addresses these concerns about the overlap in text with published work.

We will carefully review your manuscript upon resubmission, so please ensure that your revision is thorough

Additional Editor Comments (if provided):

The paper has been appreciated by the two Reviewers. However, there are some minor aspects that need to be addressed:

1. The Authors has underlined the importance of the results of the fresh transfer on day 6 compared to day 5. However, the number of transfers in day 6 was only 43. How can the authors be so sure not to have a type 1 error? The Authors may discuss this aspect. The manuscript should be also modified according to this possible bias.

2. The strategy to transfer a fresh embryo in day 6 is not so diffuse given the idea that in day 6 the endometrium is less receptive. As a matter of fact, the number of fresh transfers in day 6 is low. The Authors stated that they have corrected for centers but results are not shown. Is it possible that few centers adopted this strategy and they had older patients?

3. Was there some kind of standardization of blastocyst morphology among centers?

4. Given the small ORs, the conclusions seem overemphasized.

5. The choice of the bHCG levels is debatable. Please add some references to support this choice.

Reviewers' comments:

Reviewer's Responses to Questions

**Comments to the Author**

1. Is the manuscript technically sound, and do the data support the conclusions?

Reviewer #1: Yes

Reviewer #2: Yes

2. Has the statistical analysis been performed appropriately and rigorously? 

Reviewer #1: Yes

Reviewer #2: Yes

3. Have the authors made all data underlying the findings in their manuscript fully available?

Reviewer #1: Yes

Reviewer #2: Yes

4. Is the manuscript presented in an intelligible fashion and written in standard English?

Reviewer #1: Yes

Reviewer #2: Yes

5. Review Comments to the Author

Reviewer #1: I must congratulate you for the very interesting manuscript, which addresses the effect of paternal age on embryonic development and the chance of giving rise to a pregnancy.

The data shown will help to better understand the effect of paternal age on ART results.

Reviewer #2: Very nice paper and good numbers. It’s important to publish this kind of paper in order to have good data to explain our patients about impact of paternal age in reproductive medicine. Congratulations and hopefully will help many others physicians.

6. PLOS authors have the option to publish the peer review history of their article (what does this mean?). If published, this will include your full peer review and any attached files.

Reviewer #1: **Yes: **Javier Garcia-Ferreyra PhD

Reviewer #2: **Yes: **Alessandro Schuffner

---

## [Author Response · Author response to Decision Letter 0]

18 May 2022

Review 

PONE-D-21-32713

Is paternal age associated with transfer day, developmental stage, morphology, and initial hCG-rise of the competent blastocyst leading to live birth? a multicenter cohort study

PLOS ONE

Journal Requirements:

Reply: Thank you for the reminder. We have now ensured that our manuscript meets the requirements.

Reply: Instead of inclusion of the phrase “data not shown” we have referred to and uploaded the results as supporting tables. Please see the uploads, the references in the section of the Results and the section of Supporting information (in the main document after the references). 

3. Thank you for submitting the above manuscript to PLOS ONE. During our internal evaluation of the manuscript, we found significant text overlap between your submission and the following previously published works, some of which you are an author.

https://bmcpregnancychildbirth.biomedcentral.com/articles/10.1186/s12884-020-03348-2

https://www.fertstert.org/article/S0015-0282(20)32191-9/fulltext

Please revise the manuscript to rephrase the duplicated text, cite your sources, and provide details as to how the current manuscript advances on previous work. Please note that further consideration is dependent on the submission of a manuscript that addresses these concerns about the overlap in text with published work.

We will carefully review your manuscript upon resubmission, so please ensure that your revision is thorough

Reply: Thank you for pointing out that you have found some text overlap in two articles. 

The second article that you link to was written by our research group. The article has female age as exposure, and this manuscript that we uploaded to PLOSONE has male age as exposure. The article on female age was made parallel to the male age article. The outcomes in the two articles are the same. By examining the comparability between the two articles electronically, we can see that in the section of the Methods there are some small overlaps, which much be expected, since the same data and the same methodology was applied. For example in the sections “Treatment regimen” / ”Clinical setting”, “Outcomes” , “Covariates” and less in the section “Statical methods”. However, beyond these methodological issues, there is no overlap in the sections: “Introduction, “Results” and “Discussion”. Furthermore, we have referred to the article of female age in line, 102-104, 121, 196, 358, 363, 364, 377, 379, 381 and 385.

The first article that you link to has a very different topic of interest (Prevalence and associated factors of birth asphyxia among live births at Debre Tabor General Hospital, North Central Ethiopia). As we do not include data from the children it is difficult to see where the overlap should be, also when we read the paper. 

Reply: We have reviewed all the references and checked that none of them have been retracted.

Additional Editor Comments (if provided):

The paper has been appreciated by the two Reviewers. However, there are some minor aspects that need to be addressed:

1. The Authors has underlined the importance of the results of the fresh transfer on day 6 compared to day 5. However, the number of transfers in day 6 was only 43. How can the authors be so sure not to have a type 1 error? The Authors may discuss this aspect. The manuscript should be also modified according to this possible bias.

Reply: Unfortunately, it is by definition not possible to be sure that a type 1 error is avoided, as there will always be a risk of that type of error. The low number of day 6 transfers is reflected in the confidence interval of the adjusted OR: 1.06, CI (1.00;1.13) where the precision of the estimate is showed. We have added a sentence in the Discussion. Please see line 324-325 in the clean version.

2. The strategy to transfer a fresh embryo in day 6 is not so diffuse given the idea that in day 6 the endometrium is less receptive. As a matter of fact, the number of fresh transfers in day 6 is low. The Authors stated that they have corrected for centers but results are not shown. Is it possible that few centers adopted this strategy and they had older patients?

Reply: Thank you for that comment. We have adjusted for center by inclusion of “Clinic” in all our multivariable analyses. Please see the legends in Table 2 and Table 3.

3. Was there some kind of standardization of blastocyst morphology among centers?

Reply: Thank you for the comment. To reduce the risk of interobserver and intraobserver variation, most of the participating clinics have a standardized training set-up planned annually, which is mandatory to participate in. However, some variation may exist between the IVF laboratories, and therefore the multivariable analyses were adjusted for clinic. We added a sentence about this in the section of Materials and Methods. Please see line 166-168 in the clean version.

4. Given the small ORs, the conclusions seem overemphasized.

Reply: We highlight the adjusted association between male age and transfer day OR 1.06, CI (1.00;1.13), which means that with a one year increase in male age there is a 6% increased probability that the competent blastocyst will be transferred on day 6 compared to day 5. An assessment of whether this estimate is overemphasized is very relevant. The relevance per one year increase in male age is relatively small, but one should bear in mind that looking at the estimate over several years it may sum up to fairly high risk. Further, our result should be interpreted with caution, as OR overestimates the true risk ratio, particularly in case of frequent outcomes. We have added a sentence with that modification in the Discussion. Please see line 313-315 and line 322-324 in the clean version. Further we have modified the conclusions. Please see line 63-64 and line 391-393 in the clean version.

5. The choice of the bHCG levels is debatable. Please add some references to support this choice.

Reply: We do not entirely understand this comment. We have not chosen any beta-hCG-levels, and we do not report any cut-off values for hCG or anything similar. We merely compare the observed (standardized) hCG-levels.

---

## [Editor Report · Decision Letter 1]

15 Jun 2022

Is paternal age associated with transfer day, developmental stage, morphology, and initial hCG-rise of the competent blastocyst leading to live birth? a multicenter cohort study

PONE-D-21-32713R1

Dear Dr. Borgstrøm,

We’re pleased to inform you that your manuscript has been judged scientifically suitable for publication and will be formally accepted for publication once it meets all outstanding technical requirements.

Kind regards,

Paola Viganò

Academic Editor

PLOS ONE

Additional Editor Comments (optional):

The manuscript is acceptable for publication.
---

## [Editor Report · Acceptance letter]

30 Jun 2022

PONE-D-21-32713R1 

Is paternal age associated with transfer day, developmental stage, morphology, and initial hCG-rise of the competent blastocyst leading to live birth? a multicenter cohort study 

Dear Dr. Borgstrøm:

I'm pleased to inform you that your manuscript has been deemed suitable for publication in PLOS ONE. Congratulations! Your manuscript is now with our production department. 

Kind regards, 

on behalf of

Dr. Paola Viganò 

Academic Editor

PLOS ONE